# Evidence of pyrethroid resistance in *Anopheles amharicus* and *Anopheles arabiensis* from Arjo-Didessa irrigation scheme, Ethiopia

**Assalif Demissew**[1,2]*, **Abebe Animut**[2], **Solomon Kibret**[3], **Arega Tsegaye**[4,5],
**Dawit Hawaria**[5,6,7], **Teshome Degefa**[5,7], **Hallelujah Getachew**[5,7,8], **Ming-Chieh Lee**[3],
**Guiyun Yan**[3], **Delenasaw Yewhalaw**[5,7]

1 Department of Medical Laboratory Sciences, College of Medicine and Health Sciences, Ambo University, Ambo, Ethiopia, 2 Aklilu Lemma Institute of Pathobiology, Addis Ababa University, Addis Ababa, Ethiopia, 3 Program in Public Health, University of California at Irvine, Irvine, California, United States of America, 4 Department of Biology, College of Natural Science, Jimma University, Jimma, Ethiopia, 5 Tropical and Infectious Diseases Research Center (TIDRC), Jimma University, Jimma, Ethiopia, 6 Yirgalem Hospital Medical College, Yirgalem, Ethiopia, 7 School of Medical Laboratory Sciences, Faculty of Health Sciences, Jimma University, Jimma, Ethiopia, 8 Department of Medical Laboratory Sciences, Arbaminch College of Health Sciences, Arba Minch, Ethiopia

* assalifd@yahoo.com, ashifera2005@gmail.com

**Data Availability Statement:** All relevant data are within the manuscript.

**Funding:** This study was financially supported by National Institutes of Health (NIH) (Grant No.:

## Abstract

### Background

Indoor residual spraying and insecticide-treated nets are among the key malaria control intervention tools. However, their efficacy is declining due to the development and spread of insecticide resistant vectors. In Ethiopia, several studies reported resistance of *An. arabiensis* to multiple insecticide classes. However, such data is scarce in irrigated areas of the country where insecticides, pesticides and herbicides are intensively used. Susceptibility of *An. gambiae* s.l. to existing and new insecticides and resistance mechanisms were assessed in Arjo-Didessa sugarcane plantation area, southwestern Ethiopia.

### Methods

Adult *An. gambiae* s.l. reared from larval/pupal collections of Arjo-Didessa sugarcane irrigation area and its surrounding were tested for their susceptibility to selected insecticides. Randomly selected *An. gambiae* s.l. (dead and survived) samples were identified to species using species-specific polymerase chain reaction (PCR) and were further analyzed for the presence of knockdown resistance (*kdr*) alleles using allele-specific PCR.

### Results

Among the 214 *An. gambiae* s.l. samples analyzed by PCR, 89% (n = 190) were *An. amharicus* and 9% (n = 20) were *An. arabiensis*. Mortality rates of the *An. gambiae* s.l. exposed to deltamethrin and alphacypermethrin were 85% and 86.8%, respectively. On the other hand, mortalities against pirmiphos-methyl, bendiocarb, propoxur and clothianidin were 100%, 99%, 100% and 100%, respectively. Of those sub-samples (*An. amharicus* and *An.*

D43TW001505, R01AI050243 and U19AI129326). The funders had no role in the study design, data collection and analysis, decision to publish or preparation of the manuscript.

**Competing interests:** The authors declared that no competing interest exist.

*arabiensis*) examined for presence of *kdr* gene, none of them were found to carry the L1014F (West African) allelic mutation.

## Conclusion

*Anopheles amharicus* and *An. arabiensis* from Arjo-Didessa sugarcane irrigation area were resistant to pyrethroids which might be synergized by extensive use of agricultural chemicals. Occurrence of pyrethroid resistant malaria vectors could challenge the ongoing malaria control and elimination program in the area unless resistance management strategies are implemented. Given the resistance of *An. amharicus* to pyrethroids, its behavior and vectorial capacity should be further investigated.

## Background

Indoor residual insecticide spraying (IRS) and insecticide-treated nets (ITNs) are key strategies to prevent malaria transmission [1, 2]. However, these interventions are threatened due to the increasing occurrence of insecticide resistant malaria vectors [3–5]. Target site resistance, metabolic resistance, cuticular and behavioral resistance are among the resistance mechanisms in the vectors [4, 6, 7]. Knockdown resistance (*kdr*) is a target site resistance mechanism conferred by mutation(s) in the voltage-gated sodium channel (VGSC) gene. A single amino acid substitution of Leucine with Phenylalanine (L1014F/*Kdr*-West) and Leucine with Serine (L1014S/*Kdr*-East) at position 1014 of VGSC gene are the most common *kdr* mutations [8, 9].

The primary malaria vector in Ethiopia, *An. arabiensis*, has developed resistance to multiple insecticides such as DDT and pyrethroids, possibly due to their long term use for IRS and ITN [10–12]. Resistance in malaria vectors could be enhanced by the use of similar insecticides (chemicals) in agricultural practices [13, 14]. Small and large scale irrigation agricultural activities are increasing year after year to meet the food and economic demands of the population in Ethiopia [15, 16]. Arjo-Didessa sugarcane irrigation is one of the state owned macro-agroeconomic projects in the country [17, 18]. Insecticides, pesticides and herbicide are being extensively used for the control of malaria vectors, agricultural pests and weeds in the irrigation area and its surrounding.

Pyrethroid (deltamethrin and alpha-cypermethrin) impregnated LLINs and carbamate (propoxur and/or bendiocarb) based IRS are used to control adult malaria vectors while temephos (Abate formula) is applied for larval mosquito control in western Ethiopia including Arjo-Didessa irrigation area (Sources: Arjo-Didessa Sugar Factory malaria prevention and control department and Jimma-Arjo District Health Office). Chlorpyrifos (CPS) has been used as pesticide for the sugarcane plantation while Ametryn, Afratine and bound off (glyphosate) as herbicides against broad leaved and grass weeds in the irrigation and its surroundings (Personal communication: Local farmers and Mr. Getechew Etefa; Arjo-Didessa Sugar Factory, Agricultural Chemical's Section Head). These chemicals may enhance development of insecticide resistance through creating selection pressure [4, 19–25]. Use of agriculture pesticide has been associated with *An. gambiae* s.l. resistance in West Africa [19, 21, 26], *An. arabiensis* resistance in Sudan [27] and *An. gambiae* s.l. resistance in Tanzania [6]. High metabolic resistance of *An. arabiensis* to pyrethroids [28] and increased frequencies of *kdr* mutation were attributed to massive use of DDT and pyrethroids in cotton growing farms [19–21]. However, similar mortalities of *An. gambiae s.l* populations were observed regardless of the pesticide use pattern in areas of varying agrochemical use in Côte D'Ivoire [29].

To avert the effect of multiple and widespread insecticide resistance, clothianidin: a novel neonicotinoid insecticide, has been recommended to supplement the current insecticide based interventions [30–32]. Thus, together with the existing insecticides, it is imperative to evaluate efficacy of clothianidin against *An. gambiae* s.l. [33] in field setups including in irrigation areas.

Although several studies reported widespread resistance in *An. gambiae* s.l. most importantly in *An. arabiensis* [10–12], data on insecticide resistance and associated mechanisms is scarce in irrigated areas of Ethiopia. Furthermore, despite multiple studies conducted on resistance of *An. gambiae* s.l. (primarily *An. arabiensis*), the susceptibility status of sibling species and minor/rare species (such as *An. amharicus*) are often overlooked [34]. Except some old information on its zoophilic behavior and little importance in malaria transmission [35, 36], the current role of *An. amharicus* in malaria transmission as well as its behavior and response to public health insecticides is unknown. To our knowledge, there was no study conducted to evaluate the susceptibility status of *An. amharicus* to available insecticides in Ethiopia and beyond. Therefore, this study was the first to investigate the susceptibility of this species in western Ethiopia where *An. arabiensis* and *An. amharicus* co-exist [18]. Most importantly, regular monitoring of insecticide resistance is critical for effective resistance management especially in areas where same/similar insecticide is used for both vector control and agricultural purposes [5].

Thus, susceptibility status of *An. arabiensis* and *An. amharicus* (sibling species of the gambiae complex in Ethiopia) to commonly used insecticides and clothianidin (a new candidate insecticide), and presence of West African knockdown resistance gene (kdr-west) were investigated at Arjo-Didessa sugarcane irrigation area and its surrounding villages.

## Methods and materials

### Study area

The study was conducted at Arjo-Didessa sugarcane irrigation area and its vicinity, Oromia Region, Ethiopia (Fig 1), from September to November 2019. The study setting and socio-demographic characteristics of the inhabitants has been described elsewhere in previous studies [17, 18, 37]. Currently, the sugarcane irrigation area covers about 5,000 hectare (ha) of land with huge future expansion plans [18, 37]. The area is malarious [17, 37] where diverse *Anopheles* species including *An. arabiensis*, *An. coustani*, *An. pharoensis* and *An. amharicus* co-occur [18].

The irrigation area has been classified into 11 agricultural commands (clusters by the International Center of Excellence for Malaria Research (ICEMR) project in which this study was part). For this study, eight clusters were randomly selected for larval/pupal collection on the basis of anopheline larvae availability, larval density and habitat distribution. These were Command-2, command-3, Command-4, Command-5, Command-6 and Abote Didessa (from the irrigation scheme), and from Kerka and Didessa clusters (clusters outside the irrigation area) (Fig 1). The minimum distance between villages from the irrigation area was about 3 kilo meter.

### Anopheline larvae collection, rearing and identification

Anopheline larvae and pupae were collected from the breeding habitats of eight clusters and reared to adults in Arjo-Didessa ICEMR Insectary. The main habitat types in the irrigated clusters were manmade ponds, tyre tracks, sugarcane farm ditches, hippopotamus trenches, hoof prints of hippopotamus and seepages from irrigation canals and gate valves (Fig 2).

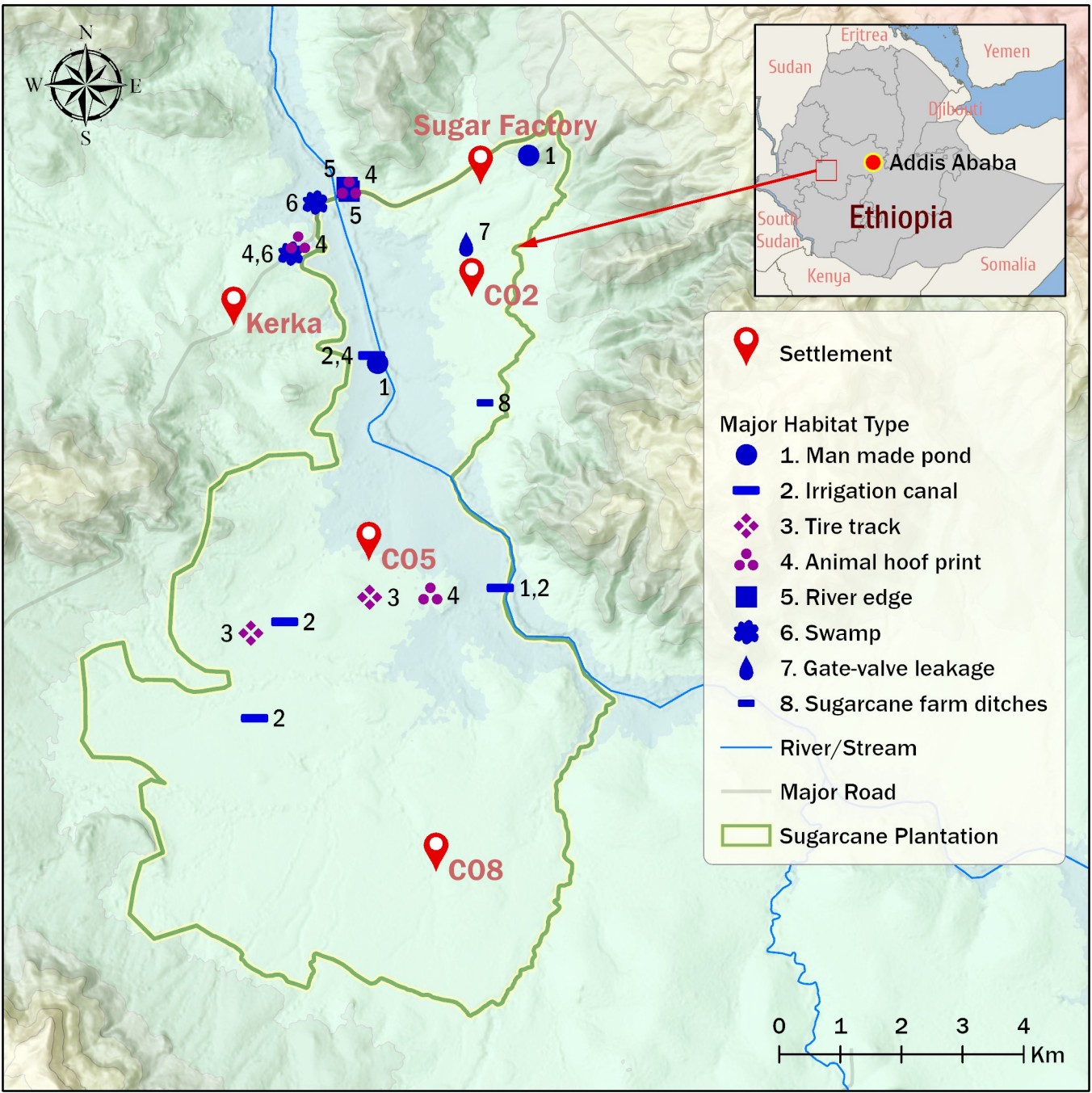

**Fig 1. Major *Anopheles* larvae collection sites, Arjo-Didessa sugarcane irrigation area and its surrounding, southwestern Ethiopia, September to November, 2019.** [This map was made using ESRI ArcGIS Pro2.8 with publicly available datasets from NASA, OpenStreetMap, and field surveys].

Whereas, the major habitats outside the irrigated clusters include animal hoof prints, stagnant water, swamps/marshes, river edges and stream seepages.

*Anopheline* larvae and pupae were collected using dipper (350 ml, Bio Quip Products, Inc. California, USA), transported to ICEMR Insectary and reared to adults in enamel trays (27×16×6.5 cm). Larvae were fed with finely ground fish food (Tetramin baby) while pupae

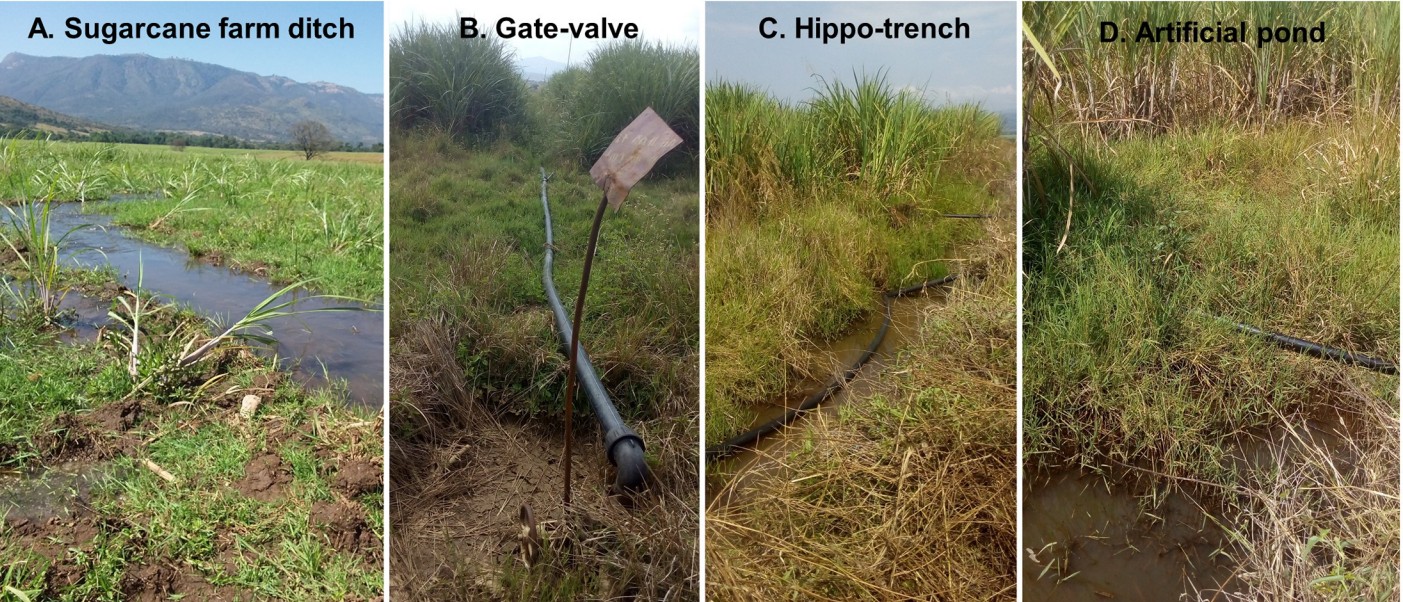

**Fig 2. Major *Anopheles* breeding habitat types in Arjo-Didessa sugarcane irrigation area, southwestern Ethiopia, September to November 2019.**

were transferred to cages and allowed to emerge to adults. The emerged adults were provided 10% sucrose solution. The mosquitoes were kept under standard conditions (25 ± 2˚C temperature, 70% ± 10% relative humidity) [38]. Adult female *Anopheles* mosquitoes were identified to species morphologically [39] and *An. gambiae* s.l. were further identified to sibling species using species-specific polymerase chain reaction (PCR) assay [18, 40].

## Insecticide susceptibility tests

Knock-down and mortality of *An. gambiae* (s.l.) females resulting from tarsal contact with insecticide impregnated papers with discriminating doses were assessed using WHO susceptibility test kits [5]. Deltamethrin (0.05%), alphacypermethrin (0.05%), bendiocarb (0.1%), propoxur (0.1%), and pirimiphos-methyl (0.25%) impregnated papers (obtained from Vector Control Research Unit, School of Biological Sciences, Malaysia) and clothianidin (2%) impregnated papers (Sumitomo chemical, Japan; Lot: CL190805) were tested for their efficacy against adult *An. gambiae* s.l. females (later identified to sibling species). In the selection of the test insecticides, current and previous insecticide usage profile of the national malaria control program was considered.

Four replicates of 24–25 non-blood fed, 3–5 days old females mosquitoes were exposed to insecticide impregnated papers for 1hour. A minimum of 99 and a maximum of 100 mosquitoes were exposed and fifty were used as a control for each insecticide tested. The number of knocked down mosquitoes were recorded at 10, 15, 20, 30, 40, 50 and finally after 60 minutes (1hour) exposure time [5]. In parallel, two replicates of mosquitoes (25x2) were exposed to control papers impregnated with silicone oil as control for pyrethroids and olive oil for organophosphate/carbamate insecticides. For clothianidin, water impregnated untreated papers obtained from similar company (Sumitomo chemical, Japan, Lot: UC190815) were used as negative controls.

After one-hour exposure time, mosquitoes were transferred back into holding tubes, provided with 10% sugar solution and the proportion of surviving and dead mosquitoes were

recorded 24 hours post exposure. However, for clothianidin tests, 10% sugar solution was changed every 12 hours and mortality was recorded daily until 100% mortality was obtained. All the tests were performed at 25°C ±2°C and 70% ± 10% relative humidity. The quality of each insecticide impregnated paper was checked on a known susceptible laboratory colony of *An. arabiensis* obtained from Sekoru insectary, Tropical and Infectious Diseases Research Centre, Jimma University (JU TIDRC), Ethiopia.

From each test, randomly selected samples of dead and surviving mosquitoes were preserved individually in Eppendorf tubes over silica-gel and kept in a freezer (-21°C) for subsequent molecular species identification and *kdr* allele detection [5].

## Identification of *An. gambiae* s.l. sibling species

About 36% (n = 214: 185 dead and 29 survived) of the 598 adult *An. gambiae* s.l samples tested for susceptibility to insecticides were identified to sibling species using polymerase chain reaction (PCR) assay following the methods of Scott *et al.* [40]. In brief: genomic DNA was extracted from legs and wings of individual mosquito using DNA extraction kit (Qiagen, Sigma Aldrich, USA). The extracted DNA product was amplified by PCR using universal primer (UN: 5'-GTGTGCCCCTTCCTCGATGT-3') and species specific primers for *An. gambiae s.s* (GA: 5'-CTGGTTTGGTCGGCACGTTT-3'), *An. arabiensis* (AR: 5'-AAGTGTCCTTCTCCA TCCTA-3') and *An. amharicus* (QD: formerly *An. quadriannulatus* B; 5'-CAGACCAAGATG GTTAGTAT-3'). Then the amplicon was loaded on 2% agarose gel stained with ethidium bromide and run for gel electrophoresis. *Anopheles arabiensis* from Sekoru Insectary colony and previously confirmed *An. amharicus* [18] were used as positive controls.

## Detection of knock down resistance gene mutation

Detection of knock down resistance gene, *kdr*-west (L1014F), mutation was carried out on 141 (n = 115 dead and n = 26 surviving) randomly selected *An. amharicus* and *An. arabiensis* (PCR identified) samples as described by Martinez-Torres *et al.* [41].

Briefly, genomic DNA extracted from individual mosquito (susceptible and resistant samples) was genotyped using allele specific primers [41]. Four allele specific primers namely Agd1 (5'-ATAGATTCCCCGACCATG-3'), Agd2 (5'-AGACAAGGATGATGAACC-3'), Agd3 (5'-AATTTGCATTACTTACGACA-3') and Agd4 (5'-CTGTAGTGATAGGAAATTTA-3') were used for the PCR amplification of *kdr*-west gene. The PCR reaction conditions were 94°C/5min x 1 cycle, (94°C/1min, 48°C/2min, 72°C/2min) x 40 cycles, 72°C/10min x 1 cycle, 4°C hold cycling condition. The amplicon was run on a 2% agarose gel and stained with ethidium bromide. Resulting fragments (bands) were interpreted as: 293pb internal control, 195bp resistant and 137bp susceptible/wild type mosquitoes [38, 41]. Susceptible *An. arabiensis* strains from Sekoru insectary colony of Jimma University, Tropical and Infectious Diseases Research Center, Ethiopia, was used as control.

## Data analysis

Data entry and analysis were made using Microsoft Excel (Version 2016, Microsoft Corp, USA) and IBM SPSS version 20.0 (SPSS Inc., Chicago, IL, USA) statistical software packages. The status of susceptibility/resistance to insecticides after 24 hours post exposure was determined using percentage mortality. Mosquitoes' phenotypic resistance status was interpreted according to WHO criteria (i.e. mortality rate $\geq$98% as susceptible; mortality rate between 90–97%, suspected/potential resistance; and mortality <90%, resistant) [38]. The $KT_{50}$ and $KT_{90}$ (time to knockdown 50% and 90% mosquitoes) values were calculated for each insecticide using log-probit analysis using SPSS v20.0 for windows statistical software.

### Ethical clearance

Ethical clearance was obtained from the Institutional Review Board (IRB) of Aklilu Lemma Institute of Pathobiology, Addis Ababa University, Ethiopia (Ref. No. *ALIPB/IRB/012/2017/ 18*). Permission was also obtained from East Wollega and Buno Bedele Zonal Health Offices, and Arjo-Didessa Sugar factory, Oromia Regional State, Ethiopia. Oral consent was taken from the interviewees.

## Results

### Mosquito species composition

Among a total of 598 *An. gambiae* s.l. mosquitoes tested for their susceptibility to different insecticides, 569 (95.15%) of them died and 29 (4.85%) survived. Of 214 randomly selected *An. gambiae* s.l. (n = 185; 35.5% of dead and n = 29; 100% of survivors) samples analyzed using species-specific PCR, about 89% (n = 190) were *An. amharicus* and the remaining 9% (n = 20) were *An. arabiensis* ([Fig 3]). In the PCR analysis, about 98% of the samples were successfully amplified.

 *Anopheles amharicus* was predominant species for every type of insecticide tested followed by *An. arabiensis* ([Table 1]). Among the *An. gambiae* s.l. sub-samples analyzed with PCR, the proportions of *An. amharicus* tested against propoxur were 100% (n = 26/26), pirmiphos-methyl 90% (n = 28/31) and clothianidin 91.5% (n = 43/47). The molecular distribution of *An. amharicus* and *An. arabiensis* tested against different insecticide classes is shown in [Table 1].

### Insecticide susceptibility status of *Anopheles gambiae* s.l.

Mortality rates of *An. gambiae* s.l. exposed to deltamethrin and alphacypermethrin impregnated papers were 85% and 86.8%, respectively ([Table 2]). On the other hand, pirmiphos-methyl induced 100% mortality, bendiocarb 99% mortality and propoxur 100% mortality. In all the control populations tested together, mortality rates were < 5%, and therefore, Abbott's

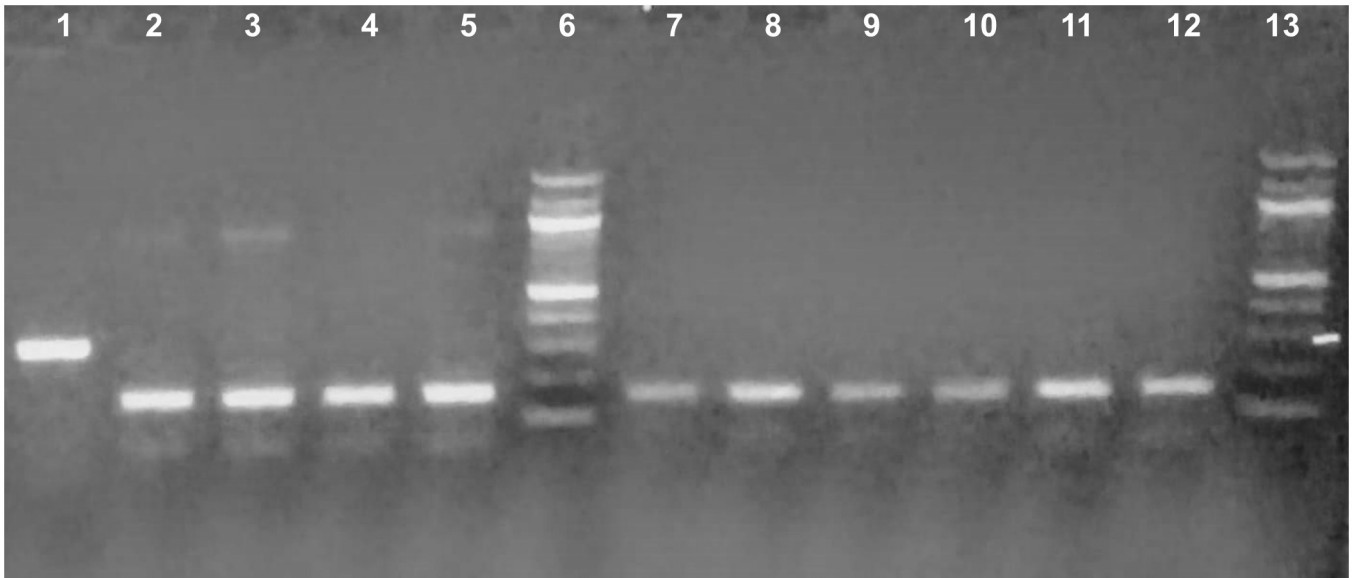

**Fig 3. Results of PCR gel electrophoresis: Lane 1; 315kb *An. arabiensis*, Lanes 2–5 & 7–12 were *An. Amharicus* (153kb); Lanes 6 and 13 were 100kb DNA ladders.**

**Table 1. Composition and insecticide susceptibility status of *An. arabiensis* and *An. amharicus* to insecticides at Arjo-Didessa irrigation scheme and its surrounding, southwestern Ethiopia, September-November, 2019.**

| Insecticide (%) | # Tested (PCR) | *An. arabiensis* | | *An. amharicus* | | UA (%) |
|---|---|---|---|---|---|---|
| | | Resistant (%) | Susceptible (%) | Resistant (%) | Susceptible (%) | |
| Deltamethrin (0.05) | 48 | 5 (10.4) | 0 (0.0) | 10 (20.8) | 33 (68.8) | 0 (0.0) |
| Alphacypermethrin (0.05) | 40 | 0 (0.0) | 0 (0.0) | 10 (25.0) | 29 (72.5) | 1 (2.5) |
| Bendiocarb (0.1) | 22 | 1 (4.5) | 8 (36.4) | 0 (0.0) | 12 (54.5) | 1 (4.5) |
| Propoxur (0.1) | 26 | 0 (0.0) | 0 (0.0) | 0 (0.0) | 25 (96.2) | 1 (3.8) |
| Pirmiphos-methyl (0.25) | 31 | 0 (0.0) | 3 (9.7) | 0 (0.0) | 28 (90.3) | 0 (0.0) |
| Clothianidin (2) | 47 | 0 (0.0) | 3 (6.4) | 0 (0.0) | 43 (91.5) | 1 (2.1) |
| Total | 214 | 6 (2.8) | 14 (6.5) | 20 (9.3) | 170 (79.4) | 4 (1.9) |

**UA: Unamplified**; three resistant & one susceptible samples

formula to correct mortality rate was not necessary during data analysis. *Anopheles arabiensis* controls from Sekoru insectary colony, JU TIDRC, were susceptible to each test insecticide used.

## Knockdown time$_{50}$ and knockdown time$_{90}$ values

The KT$_{50}$ and KT$_{90}$ values for deltamethrin were 20.6 and 80.6 minutes, respectively and the corresponding values for alphacypermethrin were 14.4 and 44.7 minutes, respectively (Table 3). The number of mosquitoes knocked down after 60-minute exposure times were eighty-four for deltamethrin and ninety-six for alphacypermethrin.

## Susceptibility status of *An. gambiae* s.l. to clothianidin

Clothianidin induced 48% (n = 48/100) knockdown after 60minutes exposure time and 94% (n = 94/100) mortality effects 24 hour post exposure against the wild *An. gambiae* s.l. The field collected *An. gambiae* s.l. (later identified as *An. amharicus* and *An. arabiensis*) exposed to clothianidin reached 100% mortality within 48 hours (2 days) post exposure while *An. arabiensis* from Sekoru insectary colony took 96 hours (4 days) to reach 100% mortality (Fig 4). From 47 *An. gambiae* s.l. sub-samples analyzed with species specific PCR assay, 91.5% (n = 43) were identified as *An. amharicus* while only 6.4% (n = 3) were *An. arabiensis*.

**Table 2. Susceptibility status of *Anopheles gambiae* s.l. to insecticides in Arjo-Didessa sugarcane irrigation area, southwestern Ethiopia, September-November, 2019.**

| Insecticide (DC) | Insecticide class | Number exposed (n) | Number dead (n) | Mortality (%) | Susceptibility | Interpretation |
|---|---|---|---|---|---|---|
| Deltamethrin (0.05%) | Pyrethroid | 100 | 85 | 85.0 | Resistant | Confirmed |
| Alphacypermethrin (0.05%) | Pyrethroid | 99 | 86 | 86.8 | Resistant | Confirmed |
| Pirmiphos-methyl (0.25%) | Organophosphate | 100 | 100 | 100.0 | Susceptible | Confirmed |
| Bendiocarb (0.1%) | Carbamate | 100 | 99 | 99.0 | Susceptible | Confirmed |
| Propoxur (0.1%) | Carbamate | 99 | 99 | 100.0 | Susceptible | Confirmed |
| Clothianidin (2%) | Neonicotinoid | 100 | 100 | 100.0[†] | Susceptible | Confirmed |

**DC**: Discriminatory Concentration,

[†]100% mortality was recorded after 48 hours post exposure

**Table 3.  Knockdown effects of deltamethrin and alphacypermethrin against *An. arabiensis* and *An. amharicus* mosquito species, Arjo-Didessa sugarcane irrigation scheme and its surrounding, southwestern Ethiopia, 2019.**

| Insecticide (%) | Wild/field mosquitoes (*An. arabiensis* & *An. amharicus*) | | Insectary colony (*An. arabiensis*) | | KT Ratio (Wild vs Colony) | |
|---|---|---|---|---|---|---|
| | $KT_{50}^*$ (95% CI) | $KT_{90}^*$ (95% CI) | $KT_{50}^*$ (95% CI) | $KT_{90}^*$ (95% CI) | $KT_{50}$ | $KT_{90}$ |
| Deltamethrin (0.05) | 20.6 (18.2–23.0) | 80.6 (66.1–106.0) | 16.34 (3.03–29.8) | 36.51 (17.75–89.62) | 1.26 | 2.21 |
| Alphacypermethrin (0.05) | 14.4 (12.5–16.1) | 44.7 (39.1–53.1) | 14.91 (1.14–29.89) | 35.17 (13.18–97.51) | 0.97 | 1.27 |

**KT**: Knockdown time; **CI**: Confidence interval;

*Time is in minute

### Detection of *kdr* (L1014F) gene mutation in *An. arabiensis* and *An. amharicus*

A total of 141 PCR confirmed (124 *An. amharicus* and 27 *An. arabiensis*) samples from dead (n = 115) and survived (n = 26) mosquitoes were examined for the occurrence of L1014F (West African *kdr*) allelic mutation. Among these, 97.2% (n = 137) were successfully amplified while only 2.8% (n = 4) were unamplified. Of those samples analyzed for presence of *kdr* gene, none of them were positive for *kdr* (L1014F) allele mutation.

## Discussion

The present study revealed the susceptibility status of *An. amharicus* and *An. arabiensis* to existing and new insecticide classes of public health use at national and continental level. Of the *An. gambiae* s.l. sub-population collected and tested from the irrigation fields and its

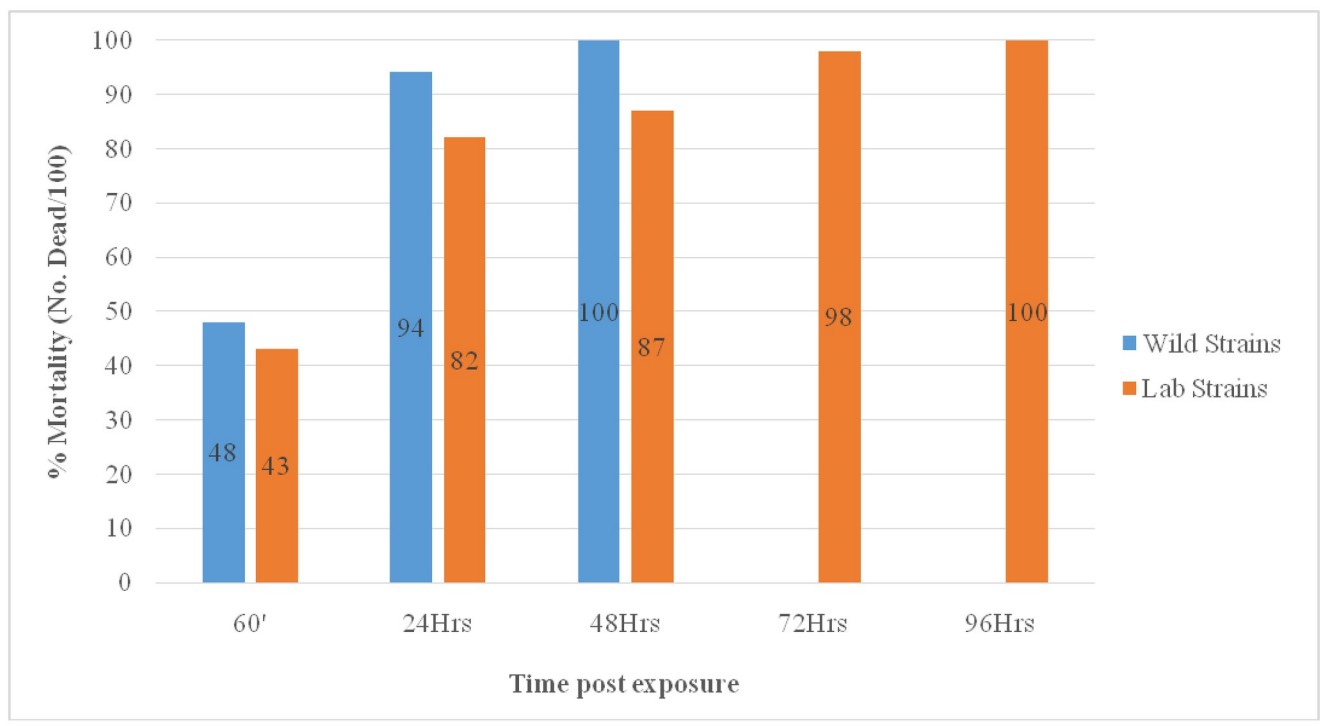

**Fig 4.  Mortality rates of the *An. gambiae* s.l. (field vs laboratory spp.) exposed to clothianidin, Arjo-Didessa sugarcane irrigation scheme and its surrounding, Ethiopia, September-November, 2019.**

surrounding, the main species identified was *An. amharicus* (~89%) while few (9%) were identified as *An. arabiensis*. Previous study from the same area also revealed the co-existence of these two sibling species in the sugarcane irrigation settings although *An. arabiensis* was the predominating species [18]. This study is the first to detect pyrethroid resistance and characterize resistance mechanisms in *An. amharicus* and *An. arabiensis* particularly in areas of agrochemical use for sugarcane irrigation activities in Ethiopia.

The dominant *An. amharicus* and *An. arabiensis* in the Arjo-Didessa sugarcane irrigation area and its surrounding villages were resistant to deltamethrin and alphacypermethrin. This could challenge the ongoing malaria vector control and elimination program as these chemicals are among the intensively used insecticides for public health and agricultural purposes in the area. This is in agreement with studies from northwestern [10], central, southwestern [10, 11, 42] and other parts of Ethiopia [12, 43] which documented resistance of *An. arabiensis* against deltamethrin. Although the mortality rates against deltamethrin (85%) and alphacypermethrin (86.8%) were comparable with a study in Gorgora, Ethiopia [10], it was relatively higher than studies in other parts of the country (were 9–75% mortalities) [10, 11, 43]. This could be attributed to the higher abundance of *An. amharicus* in our study population. Knock down time ($KT_{50}$ and $KT_{90}$) were elevated (e.g. for deltamethrin, $KT_{90}$ ratio = 2.2 compared to Lab strains) which indirectly indicate the reduced efficacy of their rapid knockdown effect which was in agreement with a study in Gorgora [10] but contrasted with a study around Gilgel-Gibe hydroelectric dam, Ethiopia [11] which could be attributed to the mixed population of the present study.

Long term use of public health insecticides and agricultural chemicals (pesticides, herbicides and fertilizers) might have contributed for the resistance of *An. arabiensis* and *An. amharicus* against pyrethroids in the study area. Studies from cotton growing areas in Burkina Faso [44] and Northern Benin [21], West Africa and southern Côte d'Ivoire [45] suggested that agricultural use of pesticides selects pyrethroid resistance within *An. gambiae* s.l. populations. In Tanzanian agricultural settings where *An. arabiensis* was the predominant species, a significant correlation was found between adult mosquitoes resistance to deltamethrin and pesticide use for agricultural activity [46]. Furthermore, high resistance of *An. gambiae* s.l. populations to pyrethroids was observed from Okyereko rice irrigation site of Ghana [14] similar to our finding. These mosquitoes exposed to sub-lethal doses of herbicides, pesticides, fertilizers or pollutants at their larval stage become more tolerant to insecticides (e.g. pyrethroides) possibly due to over expression of their detoxifying enzymes and selection of resistance genes [22, 44, 47, 48]. The interaction between xenobiotics (e.g. herbicides such as glyphosate and atrazine) present in mosquito breeding sites and the expression of mosquito genes encoding detoxification enzymes could exert the selection pressure [22, 49, 50]. Supporting our finding, a study by Oliver and Brooke, [51] demonstrated that larval exposure to glyphosate (herbicide) induced insecticide resistance in the major malaria vector, *An. arabiensis*. There was about 4.7 fold increase in deltamethrin tolerance among adult *An. arabiensis* with fertilizer exposure at their larval stage which can also be translated to an increase in pyrethroid resistance intensity due to fertilizer use [52].

In spite of the observation of phenotypic resistance to deltamethrin and alphacypermethrin, there was no *kdr* gene mutation (*kdr-w*/L1014F allele) detected among *An. arabiensis* and *An. amharicus*. However, this cannot rule out a potential involvement of other resistance mechanisms (target site, metabolic or cuticular) in the study area. For example, N1575Y mutation has been recently emerged within domains III-IV of VGSC of pyrethroid resistant *An. gambiae* population [53, 54] which might be considered as possible target site resistance mechanism in the area. Involvement of enzymatic mechanisms had also been reported in western Kenya agro-ecosystems where there were pyrethroid resistant *An. gambiae* s.l. (*An. arabiensis*) but

without *kdr* allele detection [55]. Elevation of monooxygenases and esterases enzymatic activities were observed in those resistant *An. gambiae* mosquitoes exposed to permethrin and deltamethrin [55]. A more recent study further strengthens that *An. arabiensis* with increased phenotypic resistance to pyrethroids was found with lowest *kdr*-w allelic frequency [56]. The increased number of *An. amharicus* in our test population; a species with previously unknown *kdr* frequency and with less insecticide exposure due to its exophilic and zoophagic (prefer animal shelters) behavior [57, 58], might contribute for the absence of *kdr* mutation in our study.

*Anopheles arabiensis* and *An. amharicus* were susceptible to propoxur, bendiocarb and pirimiphos-methyl insecticides. This is supported by a previous nationwide study that documented susceptibility of *An. arabiensis* to these insecticides [12] and a study in southwestern Ethiopia, susceptible to propoxur [11]. Unlike our study, bendiocarb resistance was detected in malaria vectors from Ethiopia [12] as well as rice irrigation areas of southern Côte d'Ivoire [45]. Similarly, the mosquito population were fully susceptible to the new novel insecticide, clothianidin (2%), with 100% mortality. However, 100% mortality was achieved within 48 hours (2 days) post exposure for the field strains while 96 hours (4 days) to reach 100% mortality for the laboratory strains. A similar trend was reported by a study in Ethiopia where field population of *An. arabiensis* was more susceptible to clothianidin reaching 100% mortality by day two compared to the laboratory strain reaching 100% mortality by day three [32]. Such increased susceptibility in the field strains may result from fitness cost due to presence of resistance and cross-resistance traits found in the wild populations [32, 59]. Similar studies in Ethiopia [32] and Africa [60] reported clothianidin susceptibility of malaria vectors. From its efficacy, clothianidin is being highly recommend as viable candidate to replace the current insecticides used in IRS for the control of insecticide resistant malaria vectors [30, 61, 62]. Therefore, clothianidin can be utilized as alternative/supplement for malaria vector control and elimination operations by National Malaria Elimination Program in Ethiopia.

The findings from this study strongly suggest for implementation of inter-sectoral integrated insecticide resistance management strategies involving public health, agricultural and environmental sectors, by incorporating novel chemicals such as clothianidin. This could help to reduce insecticide resistance in malaria vectors at such irrigation settings.

## Limitations of the study

Although target site (*kdr*-w) resistance mechanism was investigated, other mechanisms such as metabolic, cuticular and behavioral resistances were not determined in this study. This calls for the need for further investigations in these areas. In addition, insecticide resistance status of mosquitoes from non-irrigated (control) villages was not determined due to the critical shortage of positive larval habitats for the number of bioassay tests during the study period.

## Conclusion and recommendations

In the Arjo-Didessa sugarcane irrigation area and its surrounding villages, *An. amharicus* (for the first time) and *An. arabiensis* were observed to be resistant to pyrethroid insecticides. This brings additional challenge on current malaria vector control programs in the irrigation areas. Integrated resistance management strategies are critically needed to improve malaria vector control. Susceptibility of the study population against carbamates and organophosphate insecticides could help to exploit them as alternative chemicals for insecticide resistance management. Given the resistance of *An. amharicus* to pyrethroids, its behavior, blood feeding pattern and vectorial capacity should be further investigated. Although *kdr*-w gene mutation was not detected in our study, other resistance mechanisms including *kdr*-e should not be ruled out.

## Supporting information

**S1 Raw images. Uncropped (raw) image of gel electrophoresis result (blot).**
(PDF)

## Acknowledgments

We would like to acknowledge Arjo-Didessa sugar factory and the surround community for their cooperation while conducting this study. We are very thankful for the ICEMR field Entomology data collectors for their help in collection and rearing of mosquitoes. We are also very grateful to the Laboratory staffs; with special thanks to Mr Kasahun Zeleke and Ms Mebrat Kiya at Sekoru Parasitology and Molecular Biology Laboratory, TIDRC, Jimma University for their technical support while performing molecular analysis.

## Author Contributions

**Conceptualization:** Assalif Demissew, Abebe Animut, Guiyun Yan, Delenasaw Yewhalaw.

**Data curation:** Assalif Demissew, Abebe Animut, Arega Tsegaye, Teshome Degefa, Hallelujah Getachew, Ming-Chieh Lee, Delenasaw Yewhalaw.

**Formal analysis:** Assalif Demissew, Abebe Animut, Solomon Kibret, Arega Tsegaye, Dawit Hawaria, Teshome Degefa, Hallelujah Getachew, Ming-Chieh Lee, Guiyun Yan, Delenasaw Yewhalaw.

**Funding acquisition:** Solomon Kibret, Ming-Chieh Lee, Guiyun Yan, Delenasaw Yewhalaw.

**Investigation:** Assalif Demissew, Abebe Animut, Solomon Kibret, Arega Tsegaye, Dawit Hawaria, Teshome Degefa, Hallelujah Getachew, Guiyun Yan, Delenasaw Yewhalaw.

**Methodology:** Assalif Demissew, Abebe Animut, Solomon Kibret, Dawit Hawaria, Hallelujah Getachew, Guiyun Yan, Delenasaw Yewhalaw.

**Project administration:** Assalif Demissew, Solomon Kibret, Ming-Chieh Lee, Guiyun Yan, Delenasaw Yewhalaw.

**Resources:** Solomon Kibret, Ming-Chieh Lee, Guiyun Yan, Delenasaw Yewhalaw.

**Software:** Assalif Demissew, Ming-Chieh Lee.

**Supervision:** Assalif Demissew, Solomon Kibret, Teshome Degefa, Ming-Chieh Lee, Guiyun Yan, Delenasaw Yewhalaw.

**Validation:** Assalif Demissew, Guiyun Yan, Delenasaw Yewhalaw.

**Visualization:** Assalif Demissew, Guiyun Yan, Delenasaw Yewhalaw.

**Writing – original draft:** Assalif Demissew.

**Writing – review & editing:** Assalif Demissew, Abebe Animut, Solomon Kibret, Arega Tsegaye, Dawit Hawaria, Teshome Degefa, Hallelujah Getachew, Ming-Chieh Lee, Guiyun Yan, Delenasaw Yewhalaw.

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
