## [Decision Letter · Decision Letter 0]

20 Oct 2021

PONE-D-21-30434Evidence of pyrethroid resistance in Anopheles amharicus and Anopheles arabiensis from Arjo-Didessa irrigation scheme, EthiopiaPLOS ONE

Dear Dr. Demissew,

Thank you for submitting your manuscript to PLOS ONE. After careful consideration, we feel that it has merit but does not fully meet PLOS ONE’s publication criteria as it currently stands. Therefore, we invite you to submit a revised version of the manuscript that addresses the points raised during the review process. Please submit your revised manuscript by Dec 04 2021 11:59PM. If you will need more time than this to complete your revisions, please reply to this message or contact the journal office at plosone@plos.org. Please include the following items when submitting your revised manuscript:A rebuttal letter that responds to each point raised by the academic editor and reviewer(s). You should upload this letter as a separate file labeled 'Response to Reviewers'.A marked-up copy of your manuscript that highlights changes made to the original version. You should upload this as a separate file labeled 'Revised Manuscript with Track Changes'.An unmarked version of your revised paper without tracked changes. You should upload this as a separate file labeled 'Manuscript'.

We look forward to receiving your revised manuscript.

Kind regards,

Xinghui Qiu, Ph.D

Academic Editor

PLOS ONE

Journal Requirements:

3. We note that Figure 1 in your submission contain map/satellite images which may be copyrighted. All PLOS content is published under the Creative Commons Attribution License (CC BY 4.0), which means that the manuscript, images, and Supporting Information files will be freely available online, and any third party is permitted to access, download, copy, distribute, and use these materials in any way, even commercially, with proper attribution. For these reasons, we cannot publish previously copyrighted maps or satellite images created using proprietary data, such as Google software (Google Maps, Street View, and Earth). For more information, see our copyright guidelines: http://journals.plos.org/plosone/s/licenses-and-copyright.

a) You may seek permission from the original copyright holder of Figure 1 to publish the content specifically under the CC BY 4.0 license.  

Reviewers' comments:

Reviewer's Responses to Questions

**Comments to the Author**

1. Is the manuscript technically sound, and do the data support the conclusions?

Reviewer #1: Yes

Reviewer #2: Yes

2. Has the statistical analysis been performed appropriately and rigorously? 

Reviewer #1: Yes

Reviewer #2: Yes

3. Have the authors made all data underlying the findings in their manuscript fully available?

Reviewer #1: Yes

Reviewer #2: Yes

4. Is the manuscript presented in an intelligible fashion and written in standard English?

Reviewer #1: Yes

Reviewer #2: Yes

5. Review Comments to the Author

Reviewer #1: Review manuscript PONE-D-21-30434

Indoor residual spraying and insecticide-treated nets helped to significantly reduce the malaria burden. However, their efficacy is declining due to the development and spread of insecticide resistant vectors which is causing by massive use of insecticides in vector control and selection from agricultural pesticides. It is there important to monitoez the extent of resistance in malaria vector in order to implement suitable resistance management strategies

In this study, The evaluated the Susceptibility of An. gambiae s.l. to existing and new insecticides and resistance mechanisms in ArjoDidessa sugarcane plantation area, southwestern Ethiopia.

. They found that mosquitoes collected were susceptible to pyrethroids (deltamethrin and alphacypermethrin) but suceptible to all other classes tested. They did not detected the presence of Kdr W

The results obtained can guide in option for resistance management for example, the Susceptibility of the study population against carbamates and organophosphate insecticides could help to exploit them as alternative chemicals for insecticide resistance management. However, I have a major concerning the some of the aspect of this work

- Why monitoring the distribution of Kdrw while Ethiopia is in the eastern part where we expected to have more kdrE (the L1014S mutation). This need to be investigated

- Why testing two pyrethroids type II. It will be more informative to tested at least one type I pyrethroid such as permethrin

- The synergistic testing with PBO is missing which can inform about potential resistance mechanisms involved in the observed resistance

- The authors need to evaluate the intensity of resistance using the 5x and 10x concentrations of insecticides as recommended by WHO

- Having the results of the bioefficacy of bed nets could have provide more information about the impact of the observed resistance on the efficacy of control tools

Reviewer #2: See comments in the attached file. In general the manuscripts addresses an important is vector control issue withe reference to An amuharicus. The authors should indicate the difference in transmission capacity between An. amuharicus and An. arabiensis.

6. PLOS authors have the option to publish the peer review history of their article (what does this mean?). If published, this will include your full peer review and any attached files.

Reviewer #1: No

Reviewer #2: **Yes: **Dr. Andrew K. Githeko PhD

---

## [Author Response · Author response to Decision Letter 0]

5 Nov 2021

Author’s Responses to Reviewers and Editor Comments

Title: Evidence of pyrethroid resistance in Anopheles amharicus and Anopheles arabiensis from Arjo-Didessa irrigation scheme, Ethiopia. PONE-D-21-30434

Authors list:

Assalif Demissew (assalid@yahoo.com),

Abebe Animut (animut2004@yahoo.com), 

Solomon Kibret (s.kibret@gmail.com), 

Arega Tsegaye (2003arega@gmail.com), 

Dawit Hawaria (hawaria.dawit@gmail.com)

Teshome Degefa (teshedege@gmail.com),

Hallelujah Getachew (hgetachew4@gmail.com), 

Ming‑Chieh Lee (mingchil@uci.edu), 

Guiyun Yan (guiyuny@hs.uci.edu), 

Delenasaw Yewhalaw (delenasawye@yahoo.com)

Author’s response to reviewers:

Editor comments on journal requirements: when submitting your revision, we need you to address these additional requirements.

Author’s Response: Dear editor in chief, we are very grateful for your academic edits and fast response. We prepared the manuscript according to the PLOS ONE style using the guideline/s given. Please look at the clean version of our manuscript for the format and editing requirments.

Editor’s comments:

Author’s Response: Thank you. We submitted the figures separately on “Figure Submission” section and the original blot/gel image as supporting information; labeled as S1_raw_image. [Supporting information section: Page 29: Line#: 590-591]. 

We also indicated in our cover letter that the blot/gel image data is in the supporting information.

Editor’s comments:

3. We note that Figure 1 in your submission contain map/satellite images which may be copyrighted. All PLOS content is published under the Creative Commons Attribution License (CC BY 4.0), which means that the manuscript, images, and Supporting Information files will be freely available online, and any third party is permitted to access, download, copy, distribute, and use these materials in any way, even commercially, with proper attribution. For these reasons, we cannot publish previously copyrighted maps or satellite images created using proprietary data, such as Google software (Google Maps, Street View, and Earth). For more information, see our copyright guidelines: http://journals.plos.org/plosone/s/licenses-and-copyright.

 a) You may seek permission from the original copyright holder of Figure 1 to publish the content specifically under the CC BY 4.0 license. 

 We recommend that you contact the original copyright holder with the Content Permission Form (http://journals.plos.org/plosone/s/file?id=7c09/content-permission-form.pdf) and the following text: “I request permission for the open-access journal PLOS ONE to publish XXX under the Creative Commons Attribution License (CCAL) CC BY 4.0 (http://creativecommons.org/licenses/by/4.0/). Please be aware that this license allows unrestricted use and distribution, even commercially, by third parties. Please reply and provide explicit written permission to publish XXX under a CC BY license and complete the attached form.” Please upload the completed Content Permission Form or other proof of granted permissions as an "Other" file with your submission.

Author’s Response: Dear Editor, thank you for the comment. We understand and fully aware that copyright is a serious issue if one uses a product without permission. However, the figure in our manuscript is free of copyright issues. It (Figure 1) was done by one of the co-authors, Dr. Ming-Chieh Lee (mingchil@uci.edu), using ESRI ArcGIS Pro 2.8 software with publicly available datasets from NASA (SRTM DEM), OpenStreetMap (major road and river), and field surveys (settlement and habitat types). We took the geo-points (GPS) by ourselves and we prepared the map from those coordinates. Therefore, we confirm that figure 1 does not contain any proprietary data. The copyright of figure 1 does not belong to any other 3rd party. 

 Also, look at under caption of figure 1, we put the statement: “This map was made using ESRI ArcGIS Pro 2.8 with publicly available datasets from NASA, OpenStreetMap, and field surveys”. [Page 6, Line# 123-125]

Reviewer Comments to the Authors (Responses highlighted with blue and red colors)

NB: The Pages and Line numbers indicated in Author’s response are within Track Change MS.

Reviewer #1 general remarks: Review manuscript PONE-D-21-30434

Indoor residual spraying and insecticide-treated nets helped to significantly reduce the malaria burden. However, their efficacy is declining due to the development and spread of insecticide resistant vectors which is causing by massive use of insecticides in vector control and selection from agricultural pesticides. It is there important to monitoez the extent of resistance in malaria vector in order to implement suitable resistance management strategies.

In this study, the evaluated the Susceptibility of An. gambiae s.l. to existing and new insecticides and resistance mechanisms in Arjo Didessa sugarcane plantation area, southwestern Ethiopia. They found that mosquitoes collected were susceptible to pyrethroids (deltamethrin and alphacypermethrin) but suceptible to all other classes tested. They did not detect the presence of Kdr W. 

Reviewer #1 comment:

The results obtained can guide in option for resistance management for example, the Susceptibility of the study population against carbamates and organophosphate insecticides could help to exploit them as alternative chemicals for insecticide resistance management. 

However, I have a major concerning the some of the aspect of this work

1. Why monitoring the distribution of Kdrw while Ethiopia is in the eastern part where we expected to have more kdrE (the L1014S mutation). This need to be investigated

Author’s Response: we thank the reviewer for the comments, suggestions and concerns. It’s true that Kdr-E (L1014S) is distributed in East African while Kdr-W (L1014F) is for West African mosquito populations and the authors are well aware of this situation. 

In this regard, Ethiopian’s case can be considered as an exception because unlike other East African An. gambiae s.l mosquito population, KDR-E is not yet detected in the country whereas KDR-W is widely distributed among the target site resistance mechanisms. Therefore, we looked for Kdr-W on the basis of previous studies [References below] in which no study could detect KDr-E in Ethiopia (i.e. KDR-E is not yet detected in the country).

Some Evidences on this: A study by Yewhalaw et al., (2010); one of the senior author in this research, confirmed the first evidence of West African KDr (L1014F) mutation in Ethiopia/in East Africa. Similarly, another study from central, northern and south west Ethiopia by Balkew and his colleagues (2010), detected only KDr-W among the mosquitoes tested. A more recent nationwide resistance study by Messenger et al., (2017) between 2012-2016 also failed to detect Kdr-E in Ethiopia. A study by Alemayehu et al., (2017) on mapping insecticide resistance and characterization of resistance mechanisms in Anopheles arabiensis in Ethiopia could not detect KDR-E in the country. 

However, we share the reviewers’ concern because, although it is not yet detected in Ethiopia, still we cannot rule out KDr-E in the study area and we recommend this for further investigation and included in the recommendation section. [Page 21, Line#: 383];”….other resistance mechanisms including kdr-e should not be ruled out” 

References (Studies at regional and national level):

• Balkew et al. (2010). Insecticide resistance in Anopheles arabiensis (Diptera: Culicidae) from villages in central, northern and south west Ethiopia and detection of kdr mutation. Parasites Vectors 3, 40. https://doi.org/10.1186/1756-3305-3-40

• Yewhalaw et al. (2010). First Evidence of High Knockdown Resistance Frequency in Anopheles arabiensis (Diptera: Culicidae) from Ethiopia. Am J Trop Med Hyg. 83(1)

• Messenger et al. (2017) Insecticide resistance in Anopheles arabiensis from Ethiopia (2012–2016): a nationwide study for insecticide resistance monitoring. Malar J 16: 1-14)

• Alemayehu et al. (2017). Mapping insecticide resistance and characterization of resistance mechanisms in Anopheles arabiensis (Diptera: Culicidae) in Ethiopia. Parasite Vectors 10.

Reviewer #1 comment:

2.- Why testing two pyrethroids type II. It will be more informative to tested at least one type I pyrethroid such as permethrin

Author’s Response: Thank You. In the methods and materials section, we stated our insecticide selection criteria as “In the selection of the test insecticides, current and previous insecticide usage profile of national malaria control program was considered” [Page 7, Line# 157-159]. Therefore, our test insecticide selection criteria was their current and previous use (local and national operational application/utilization) in the area for either IRS or LLINs or both. PermaNet (deltamethrin) and MagNet (alphacypermethrin) are incorporated in LLINs operations as principal chemicals and pirimiphos-methyl, propoxur and bendiocarb are among the insecticides being used for IRS in Ethiopia. Due to this reason, we hypothesize that there could be a selection pressure due to utilization of the above mentioned insecticides. We also believe that there are several studies conducted on Type-I pyrethroids in the country and decisions can be made based on those studies together with our research outcome. 

Reviewer #1 comment:

3. The synergistic testing with PBO is missing which can inform about potential resistance mechanisms involved in the observed resistance 

Author’s Response: we share the reviewer’s comment because PBO could grossly suggest the possible biochemical/enzymatic resistance mechanism in the field. We didn’t conduct PBO assay at the field because of critical shortage of mosquitoes in the study setting. We took almost three months to complete this study (September to November, 2019) due to this problem. In the area, although there were positive habitats for Anopheline mosquitoes (An. caustani complex, An. funestus group and An. phoroensis), there was shortage of An. gambiae s.l mosquito larva and pupa to conduct both PBO and intensity assays. We put this on our limitation of the study as “other mechanisms such as metabolic, cuticular and behavioral resistances were not determined in this study” on [Page 20, Line#: 369-370] indicating the need of biochemical/enzymatic investigation by other researchers.

Reviewer #1 comment:

4.-The authors need to evaluate the intensity of resistance using the 5x and 10x concentrations of insecticides as recommended by WHO

Author’s Response: We thank the reviewer for the comment. We didn’t conduct the intensity assay due to mainly logistic problem (lack of 5x and 10x concentration kits) in addition to mosquito shortage. However, from our results, the population was susceptible for majority (four) of the insecticides tested and were resistant only for deltamethrin and alphacypermethrin. For these insecticides (the pyrethroids), we believe that the mosquito mortality at the diagnostic concentration (85% and 86.8% mortality) was relatively higher compared to multiple studies in the country. This might indicate that the population had less resistance intensity. We include this in the discussion section of the original manuscript [Page 17-18, Line#:303-310]. 

Reviewer #1 comment:

5.-Having the results of the bioefficacy of bed nets could have provide more information about the impact of the observed resistance on the efficacy of control tools.

Author Response: As indicated by the reviewer, information on bio-efficacy of insecticide treated bed nets could have strengthened the result. However, this was not in our objectives.

Reviewer #2: Dr. Andrew K. Githeko (PhD)

See comments in the attached file. In general, the manuscripts address an important is vector control issue with reference to An. amharicus. The authors should indicate the difference in transmission capacity between An. amharicus and An. arabiensis.

General remarks:

The manuscript addresses an important vector control operational issues. The data generated will be useful to the Ethiopian National Malaria program.

Author’s response: We thank the reviewer for the careful review and positive remarks.

Reviewer #2 comment:

1. It would be useful to provide some information on the relative importance of An. amharicus as a vector relative to An. arbiensis.

Author’s response: We thank the reviewer for this valuable suggestion. We provide information on the vectorial importance and behaviour of An. Amharicus in the background: “Except some old information on its zoophilic behavior and little importance in malaria transmission, the current role of An. amharicus in malaria transmission as well as its behavior and response to public health insecticides is unknown” [Page 4-5, Line#: 100-103]. 

Because of the lack of current information, we recommend further investigation on the vectorial capacity and behaviour of An. amharicus as: “Given the resistance of An. amharicus to pyrethroids, its behavior, blood feeding pattern and vectorial capacity should be further investigated”. [Page 21, line:380-382]

Reviewer #2 comment:

2. Indicate which of the two species is more anthropophilic and thus more likely to come into contact with insecticides indoors.

Author’s Response: As we explained in the previous response (we also indicated this in the discussion), there is old information indicating An. amahricus to be zoophilic and exophagic [Page, 19, Line#:344-345]. “An. amharicus…. less insecticide exposure due to its exophilic and zoophagic (prefer animal shelters) behaviour”. However, current evidence on its feeding and resting behaviour is required. Therefore, we put a recommendation for further study as explained in the previous response [Page 21, line:380-382]. 

Reviewer #2

Specific remarks

Anopheles amharicus was predominant species for every type of insecticide tested followed by

An. arabiensis (Table 1). The proportions of An. amharicus tested for propoxur was 100% (n=26/26), for pirmiphos-methyl 90% (n=28/31) and for clothianidin 91.5% (n=43/47). The distribution of An. amharicus and An. arabiensis tested again.

Reviewer #2 comment: 

1. Is the above sentence referring to proportion mortality? Please clarify.

Author’s Response: We thank the reviewer for the comment. We give clarification as: “Among the An. gambiae s.l. sub-samples analysed with PCR, the proportions of An. amharicus tested against propoxur were 100% (n=26/26), pirmiphos-methyl 90% (n=28/31) and clothianidin 91.5% (n=43/47)” [Page 11, Line #: 234-236]. 

Clarification on Table 1: it shows the proportions of An. amharicus and An. arabiensis among sub samples of An. gambiae s.l. analysed with PCR (the proportion of PCR confirmed An. amharicus and An. arabiensis species).

Reviewer #2 comment: 

KT50 and KT90 values

2. Spell out KT in full in the subheading.

Author’s Response: Thank you. We correct accordingly. [Page 14, Line #: 255]

Reviewer #2 comment:

Previous study from the same area also revealed the co-existence of these two sibling species in the sugarcane irrigation settings although An. arabiensis was the predominating species [18].

3. Could the species abundance in this and in the previous studies have been affected by seasonality? An gambiae and An. arabiensis abundance in sympatric populations, is influenced by seasonality elsewhere in Africa.

Author’s Response: Yes, from our previous study, species abundance was highly affected by seasonality in the area. About 86% of the total Anopheles mosquitoes were collected during the wet seasons and the remaining 14% (n = 295) collected during the dry seasons and the difference was statistically significant (χ2 = 70.423, df = 4, P < 0.001). In the wet seasons, indoor and outdoor density of An. gambiae s.l. was the highest in the irrigated clusters [18]. 

Depending on this result, we made our study period between September and November to get a good density of mosquitoes. However, we didn’t determine mosquito abundance in this study because our objective was to determine the insecticide susceptibility status of An. gambiae s.l. within the study period.

Reviewer #2 comment:

Long term use of public health insecticides and agricultural chemicals (pesticides, herbicides and 

fertilizers) might have contributed for the resistance of An. arabiensis and An. amharicus against 

pyrethroids in the study area.

4. Is there a history of the use of DDT for vector control in the study area ?. This could explain the cross resistance to pyrethroides.

Author’s Response: DDT had been used for more than four decades at the national level, and also in particular in Arjo-Didessa (the study area) as a principal part of IRS chemical. However, nationally, DDT spraying was discontinued in 2009 (nearly 12 years before) and was replaced by deltamethrin. Therefore, the selection pressure and cross resistance (with pyrethroids) might be minimum although this needs further investigation.

5. Please provide any evidence that herbicides and fertilisers are linked to insecticide resistance and in particular to pyrethroides. The authors need to discuss the mechanisms involved. The studies in west Africa (Côte d’ Ivoire ) and Tanzania did not provide the mechanism involved.

Author’s Response: We thank the reviewer for the suggestion. We include the following evidences in the discussion section: “These mosquitoes exposed to sub-lethal doses of herbicides, pesticides, fertilizers or pollutants at their larval stage become more tolerant to insecticides (e.g. pyrethroides) possibly due to over expression of their detoxifying enzymes and selection of resistance genes [Hien et al. (2017; Akogbeto et al., 2006; Nkya et al., 2013; David et al., 2010]. The interaction between xenobiotics (e.g. herbicides such as glyphosate and atrazine) present in mosquito breeding sites and the expression of mosquito genes encoding detoxification enzymes could exert the selection pressure [Poupardin et al., 2008; Riaz et al., 2009; Nkya et al., 2013]. Supporting our finding, a study by Oliver and Brooke, [Oliver and Brooke, 2018] demonstrated that larval exposure to glyphosate (herbicide) induced insecticide resistance in the major malaria vector, An. arabiensis. There was about 4.7fold increase in deltamethrin tolerance among adult An. arabiensis with fertilizer exposure at their larval stage which can also be translated to an increase in pyrethroid resistance intensity due to fertilizer use [Jeanrenaud et al., 2019]”. [Page 18, Line #: 319-329].

6. Indicate the class of insecticides for clothianidin after its first mention: neonicotinoid

Author’s Response: Thank you. We corrected accordingly after first mention in the background as: “To avert the effect of multiple and widespread insecticide resistance, clothianidin: a novel neonicotinoid insecticide, has been recommended to supplement the current insecticide based interventions” [Page 4; Line#: 91-92]. 

Similar studies in Ethiopia [32] and Africa [52] reported clothianidin susceptibility of malaria vectors. From its efficacy, clothianidin is being highly recommend as viable candidate to replace IRS for...

7. It is nor IRS that is replaced, rather it is the insecticide such as DDT, deltamethrin, etc that are being replaced.

Author’s Response: we thank the reviewer for the correction. We correct as: “From its efficacy, clothianidin is being highly recommend as viable candidate to replace the current insecticides used in IRS for the control of insecticide resistant malaria vectors” [Page 20; Line #: 360-361]

---

## [Decision Letter · Decision Letter 1]

9 Dec 2021

Evidence of pyrethroid resistance in Anopheles amharicus and Anopheles arabiensis from Arjo-Didessa irrigation scheme, Ethiopia

PONE-D-21-30434R1

Dear Dr. Demissew,

We’re pleased to inform you that your manuscript has been judged scientifically suitable for publication and will be formally accepted for publication once it meets all outstanding technical requirements.

Kind regards,

Xinghui Qiu, Ph.D

Academic Editor

PLOS ONE

Additional Editor Comments (optional):

Reviewers' comments:

Reviewer's Responses to Questions

**Comments to the Author**

1. If the authors have adequately addressed your comments raised in a previous round of review and you feel that this manuscript is now acceptable for publication, you may indicate that here to bypass the “Comments to the Author” section, enter your conflict of interest statement in the “Confidential to Editor” section, and submit your "Accept" recommendation.

Reviewer #1: All comments have been addressed

Reviewer #2: All comments have been addressed

2. Is the manuscript technically sound, and do the data support the conclusions?

Reviewer #1: Yes

Reviewer #2: Yes

3. Has the statistical analysis been performed appropriately and rigorously? 

Reviewer #1: Yes

Reviewer #2: Yes

4. Have the authors made all data underlying the findings in their manuscript fully available?

Reviewer #1: (No Response)

Reviewer #2: Yes

5. Is the manuscript presented in an intelligible fashion and written in standard English?

Reviewer #1: (No Response)

Reviewer #2: Yes

6. Review Comments to the Author

Reviewer #1: (No Response)

Reviewer #2: No further comments. The authors have responded to all the remarks adequately. It is understood that the study is based on issues arising from the ongoing vector control around the sugar irrigation area of Ethiopia rather than a fundamental study of insecticide resistance. The authors have explained that there is little data on the vdectorial capacity of An. amharicus and they have recommended further studies in this subject. The authors have acknowledged the previous use of DDT for IRS in the area, and this could patiala explain the origin of resistance to pyrethroids.

The authors have provided supporting information to their proposition that the use of herbicides and ferterlizers can select for resistance in vectors at the larval stage. Advanced water chemistry studies are require to answer this yet to be understood issue.

7. PLOS authors have the option to publish the peer review history of their article (what does this mean?). If published, this will include your full peer review and any attached files.

Reviewer #1: No

Reviewer #2: **Yes: **Dr. Andrew K. Githeko. PhD

---

## [Editor Report · Acceptance letter]

6 Jan 2022

PONE-D-21-30434R1 

Evidence of pyrethroid resistance in *Anopheles amharicus* and *Anopheles arabiensis* from Arjo-Didessa irrigation scheme, Ethiopia 

Dear Dr. Demissew:

I'm pleased to inform you that your manuscript has been deemed suitable for publication in PLOS ONE. Congratulations! Your manuscript is now with our production department. 

Kind regards, 

on behalf of

Dr. Xinghui Qiu 

Academic Editor

PLOS ONE